# GENIE: GAUSSIAN ENCODING FOR NEURAL RADIANCE FIELDS INTERACTIVE EDITING

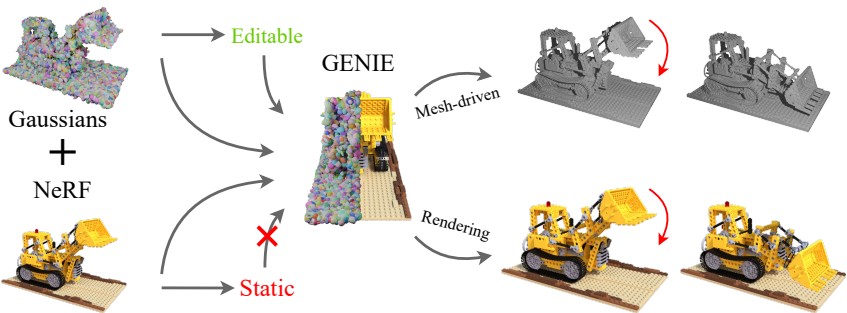

Figure 1: **GENIE capabilities.** GENIE combines the editability of Gaussians with the neural rendering power of Neural Radiance Fields (NeRF). It enables fine-grained, on-the-fly editing through either manual interaction or mesh-driven deformation.

## ABSTRACT

Neural Radiance Fields (NeRF) and Gaussian Splatting (GS) have recently transformed 3D scene representation and rendering. NeRF achieves high-fidelity novel view synthesis by learning volumetric representations through neural networks, but its implicit encoding makes editing and physical interaction challenging. In contrast, GS represents scenes as explicit collections of Gaussian primitives, enabling real-time rendering, faster training, and more intuitive manipulation. This explicit structure has made GS particularly well-suited for interactive editing and integration with physics-based simulation. In this paper, we introduce GENIE (**G**aussian **E**ncoding for **N**eural Radiance Fields **I**nteractive **E**diting), a hybrid model that combines the photorealistic rendering quality of NeRF with the editable and structured representation of GS. Instead of using spherical harmonics for appearance modeling, we assign each Gaussian a trainable feature embedding. These embeddings are used to condition a NeRF network based on the $k$ nearest Gaussians to each query point. To make this conditioning efficient, we introduce Ray-Traced Gaussian Proximity Search (RT-GPS), a fast nearest Gaussian search based on a modified ray-tracing pipeline. We also integrate a multi-resolution hash grid to initialize and update Gaussian features. Together, these components enable real-time, locality-aware editing: as Gaussian primitives are repositioned or modified, their interpolated influence is immediately reflected in the rendered output. By combining the strengths of implicit and explicit representations, GENIE supports intuitive scene manipulation, dynamic interaction, and compatibility with physical simulation, bridging the gap between geometry-based editing and neural rendering.

## 1 INTRODUCION

In recent years, we have seen significant development in the field of 3D graphics. It is primarily centered around two key tasks: the reconstruction of objects and scenes in 3D space, and the enhancement of user immersion in terms of manipulation and editing (Wang et al., 2023a; Huang et al., 2024a). Editing capabilities are essential, especially as applications in robotics, virtual en-

Figure 2: **Evolution of two physical simulations.** From left to right: (1) A rubber duck falling onto a pillow and deforming it. (2) A pirate flag waving under the influence of wind. Both simulations are performed on our own assets.

vironments, and content creation increasingly demand physically grounded simulation (Authors, 2024). Tasks like object manipulation, deformable modeling, and physics-aware animation require 3D representations that support intuitive editing and tight integration with physics engines.

In the context of scene reconstruction, neural rendering has emerged as a prominent and rapidly advancing research. A major breakthrough in this domain was the introduction of Neural Radiance Fields (NeRF) (Mildenhall et al., 2020), which transformed photogrammetry by enabling high-fidelity 3D scene reconstruction from sparse collections of 2D images and their associated camera poses. NeRFs combine neural networks with classical graphics techniques, to synthesize photorealistic views from novel perspectives. On the other hand, Gaussian Splatting (GS) (Kerbl et al., 2023) represents a recent advancement in 3D scene representation, modeling scenes as collections of Gaussian primitives with associated colour, opacity, and spatial extent.

GS employs a discrete set of Gaussians that approximate surfaces through density accumulation. This approach enables extremely fast rendering, but introduces challenges in scenarios requiring view-dependent consistency and resolution scaling (Malarz et al., 2025). In particular, when applying super-resolution or scaling transformations, gaps may appear between Gaussian components due to the inherently discrete nature of the representation. In contrast, NeRFs avoid such artifacts, making them more suitable for applications that require seamless surface continuity, such as geometry merging or fine-scale detail preservation (Mildenhall et al., 2020). Furthermore, NeRFs are typically more robust in modeling complex lighting effects and maintaining photorealistic consistency across novel viewpoints, especially under limited training data (Martin-Brualla et al., 2021).

Physics simulation enables object manipulation, collision detection, and realistic movement, which vanilla NeRF alone does not provide. Despite these needs, current NeRF representations offer limited editing capabilities. Recent works such as RIP-NeRF (Wang et al., 2023b), NeuralEditor (Chen et al., 2023), and PAPR (Zhang et al., 2023) employ 3D point clouds for conditioning. Alternatively, methods like NeRF-Editing (Yuan et al., 2022b) and NeuMesh (Yang et al., 2022) use mesh faces to control NeRF representations. In (Monnier et al., 2023), the authors model primitives as textured superquadric meshes for physics-based simulations. While these approaches introduce forms of manual editing, they remain limited in scope and are typically constrained to coarse modifications.

However, representing an object using primitives allows for direct manipulation in a manner analogous to mesh vertices, enabling intuitive, fine-grained, and real-time editing. This representation has proven highly amenable to interactive modification (Guédon & Lepetit, 2024; Waczyńska et al., 2024; Gao et al., 2025; Huang et al., 2024b), and its compatibility with physics engines (Xie et al., 2024; Borycki et al., 2024) opens the door to dynamic scene manipulation and physically grounded simulation.

In this work, we explore the potential of combining NeRF with primitive-based representations to enable object manipulation and physical simulation. This means that we can use all universal simulation methods, including highly developed external tools such as Blender (Community, 2018), to create simulations and easily assign the characteristics of a given material (plasticity, material physics). To our knowledge, no previous NeRF-based approach has demonstrated this level of integration with physical simulation frameworks, especially for large scenes. We demonstrate that our method yields superior visual and numerical results compared to existing NeRF-based methods.

In conclusion, the main contributions of this paper are as follows:

- GENIE hybrid architecture enabling the use of existing GS editing techniques for NeRF scene manipulation.

- We introduce Splash Grid Encoding, a multi-resolution encoding that conditions NeRF on spatially-selected Gaussians.
- We propose an approximate algorithm for nearest neighbor search, referred to as Ray-Traced Gaussian Proximity Search (RT-GPS) for computational overhead reduction, which enables fast and scalable inference.

## 2 RELATED WORKS

Several approaches focus on modeling deformation or displacement fields at a per-frame level (Park et al., 2021a;b; Tretschk et al., 2021; Weng et al., 2022), while others aim to capture continuous motion over time by learning time-dependent 3D flow fields (Du et al., 2021; Gao et al., 2021; Guo et al., 2023; Cao & Johnson, 2023).

A substantial body of research has also explored NeRF-based scene editing across different application domains. This includes methods driven by semantic segmentation or labels (Bao et al., 2023; Dong & Wang, 2023; Haque et al., 2023; Mikaeili et al., 2023; Song et al., 2023; Wang et al., 2022), as well as techniques that enable relighting and texture modification through shading cues (Gong et al., 2023; Liu et al., 2021; Rudnev et al., 2022; Srinivasan et al., 2021). Other efforts support structural changes in the scene, such as inserting or removing objects (Kobayashi et al., 2022; Lazova et al., 2023; Weder et al., 2023), while some are tailored specifically for facial editing (Hwang et al., 2023; Jiang et al., 2022; Sun et al., 2022) or physics-based manipulation from video sequences (Hofherr et al., 2023; Qiao et al., 2022) Geometry editing within the NeRF framework has received considerable attention (Kania et al., 2022; Yuan et al., 2022a; 2023; Zheng et al., 2023).

Our model uses geometry editing and physics simulations. Existing methods leverage various geometric primitives for conditioning NeRFs, most notably 3D point clouds. For instance, RIP-NeRF (Wang et al., 2023b) introduces a rotation-invariant point-based representation that enables fine-grained editing and cross-scene compositing by decoupling the neural field from explicit geometry. NeuralEditor (Chen et al., 2023) adopts a point cloud as the structural backbone and proposes a voxel-guided rendering scheme to facilitate precise shape deformation and scene morphing. Similarly, PAPR (Zhang et al., 2023) learns a parsimonious set of scene-representative points enriched with learned features and influence scores, enabling geometry editing and appearance manipulation.

Some approaches leverage explicit mesh representations to enable NeRF editing. NeRF-Editing (Yuan et al., 2022b) extracts a mesh from the scene and allows users to apply traditional mesh deformations, which are then transferred to the implicit radiance field by bending camera rays through a proxy tetrahedral mesh. Similarly, NeuMesh (Yang et al., 2022) encodes disentangled geometry and texture features at mesh vertices, enabling mesh-guided geometry editing and texture manipulation. To reduce computational complexity, some approaches rely on simplified geometry proxies, such as coarse meshes paired with cage-based deformation techniques (Jambon et al., 2023; Peng et al., 2022; Xu & Harada, 2022). VolTeMorph (Garbin et al., 2024) introduces an explicit volume deformation technique that supports realistic extrapolation and can be edited using standard software, enabling applications such as physics-based object deformation and avatar animation. PIE-NeRF (Feng et al., 2024) integrates physics-based, meshless simulations directly with NeRF representations, enabling interactive and realistic animations.

All of the aforementioned approaches support manual editing through explicit conditioning representations. In contrast, our method leverages a GS-based representation, allowing seamless integration with existing GS editing tools to manipulate NeRF outputs.

## 3 PRELIMINARY

Our method, GENIE, builds on two foundational models: Neural Radiance Fields (NeRF) and Gaussian Splatting (GS). We briefly review both in the following part.

**Neural Radiance Fields** Vanilla NeRF (Mildenhall et al., 2020) represents a 3D scene as a continuous volumetric field by learning a function that maps a spatial location $\mathbf{x} = (x, y, z)$ and a viewing direction $\mathbf{d} = (\theta, \psi)$, to an emitted colour $\mathbf{c} = (r, g, b)$ and a volume density $\sigma$. Formally, the scene

Figure 3: **Examples of physical simulations.** From top to bottom: (1) Rigid body simulation of falling leaves from the *NeRF Synthetic* Ficus plant. (2) Soft body simulation deforming the *NeRF Synthetic* Lego dozer. (3) Cloth simulation of fabric falling onto a cup from our custom asset collection. The middle column shows the driving mesh deformations.

is approximated by a multi-layer perceptron (MLP):

$$\mathcal{F}_{\text{NeRF}}(\mathbf{x}, \mathbf{d}; \Theta) = (\mathbf{c}, \sigma),$$

where $\Theta$ denotes the trainable network parameters.

The model is trained using a set of posed images by casting rays from each camera pixel into the scene and accumulating colour and opacity along each ray based on volumetric rendering principles. The goal is to minimize the difference between the rendered and ground-truth images, allowing the MLP to implicitly encode both the geometry and appearance of the 3D scene. To improve scalability and spatial precision, many NeRF variants adopt the Hash Grid Encoding (Müller et al., 2022), which captures high-frequency scene details by dividing space into multiple Levels of Detail (LoD), each with trainable parameters $\Phi$ and feature vectors $F$. These levels vary in resolution, allowing the encoding to represent both coarse and fine details. For a query point $\mathbf{x}$, the output feature vector $\mathbf{v}$ is obtained by concatenating trilinearly interpolated features from all levels, based on $\mathbf{x}$'s position within the grid

$$\mathcal{H}_{\text{enc}}(\mathbf{x}; \Phi) = \mathbf{v}(\mathbf{x}).$$

**Gaussian Splatting** The GS technique models a 3D scene as a set of three-dimensional Gaussian primitives. Each Gaussian is defined by a centroid position, a covariance matrix, an opacity scalar, and colour information encoded via spherical harmonics (SH) (Kerbl et al., 2023). This method builds a radiance field by iteratively optimizing the Gaussian parameters: position, covariance, opacity, and SH colour coefficients. The efficiency of GS largely stems from its rendering process, which projects these Gaussian components onto the image plane.

Formally, the scene is represented by a dense collection of Gaussians:

$$\mathcal{G}_{GS} = \left\{ (\mathcal{N}(\boldsymbol{\mu}_i, \boldsymbol{\Sigma}_i), \sigma_i, \mathbf{c}_i) \right\}_{i=1}^{n},$$

where $\boldsymbol{\mu}_i$ is the centroid location, $\boldsymbol{\Sigma}_i$ is the covariance matrix capturing anisotropic shape, $\sigma_i$ denotes opacity, and $\mathbf{c}_i$ contains the SH colour coefficients of the $i$-th Gaussian.

The optimization alternates between rendering images from the current Gaussian parameters and comparing them to the corresponding training views.

Gaussian Splatting can be easily modified in a mesh-based fashion (Guédon & Lepetit, 2024; Waczyńska et al., 2024; Gao et al., 2025; Huang et al., 2024b). In practice, this involves moving the Gaussian components directly in the 3D space.

## 4 PROPOSED METHOD

Our model, called GENIE, integrates a Gaussian representation of a shape and a neural network-based rendering procedure into a single system. Specifically, we use a set of Gaussian components $\mathcal{G}_{GS}$, where we replace the original colour vector $\mathbf{c}$ with a trainable latent feature vector $\mathbf{v} \in \mathbb{R}^n$, similar to the approach in (Govindarajan et al., 2024). We refer to this modified set of Gaussians as

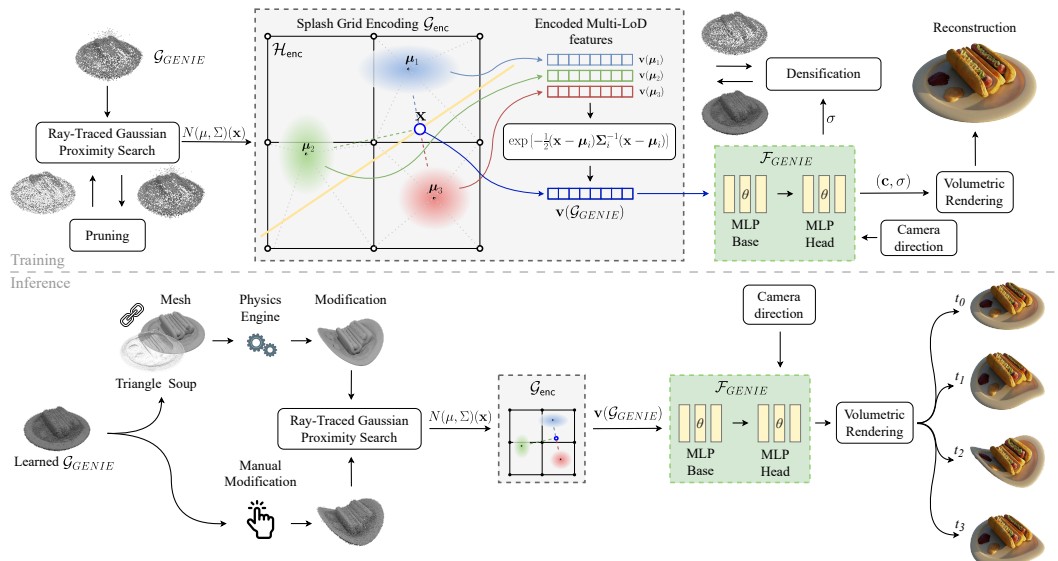

Figure 4: **Model overview.** Top: During training, a subset of Gaussians is selected using Ray-Traced Gaussian Proximity Search (RT-GPS), which also handles pruning based on Gaussian confidence. The selected Gaussians are passed to Splash Grid Encoding, which interpolates their features and drives the densification process by inserting new Gaussians as needed. The interpolated features are then processed by the neural network $\mathcal{F}_{GENIE}$ to predict colour $\mathbf{c}$ and opacity $\sigma$, which are used for volumetric rendering. Bottom: At inference, the learned Gaussians serve as input and can undergo manual or physics-driven edits. The modified Gaussians are passed through the same rendering pipeline to produce the final image.

$\mathcal{G}_{GENIE}$. To allow efficient training of anisotropic Gaussians, we adopt the standard factorization $\Sigma = RSS^T R^T$, where $R$ is a rotation matrix and $S$ is a diagonal scale matrix.

We use a NeRF-based neural network $\mathcal{F}_{GENIE}$ to predict colour and opacity from the nearest Gaussian features. Formally, the model is defined as:

$$GENIE(\mathbf{x}, \mathbf{d}; \mathcal{G}_{GENIE}, \Theta, \Phi) =$$
$$= \mathcal{F}_{GENIE}(\mathcal{G}_{enc}(\text{RT-GPS}(\mathbf{x}, \mathcal{G}_{GENIE})), \mathbf{d}) = (\mathbf{c}, \sigma),$$

where $\Theta$ and $\Phi$ denote the trainable network parameters. The model, alongside the standard NeRF input, takes a set of trainable Gaussians $\mathcal{G}_{GENIE}$ and outputs colour $\mathbf{c}$ and density $\sigma$ at any query point, enabling neural rendering conditioned on nearby Gaussian features.

**Splash Grid Encoding** The *Hash Grid Encoding*, although effective for encoding static scenes, does not support meaningful modifications. This is because altering the grid at lower LoD affects the resulting feature differently than modifying the higher-resolution levels. Consequently, editing the scene becomes inconsistent and unintuitive. To address this, we propose Splash Grid Encoding, a novel encoding mechanism that decouples feature representation from grid vertices and instead ties it to a set of Gaussians. Our method takes as input a set of query points $\mathbf{x}$ and a set of Gaussians $\mathcal{G}_{GENIE}$, and produces multi-LoD features. Formally, we define this encoding as:

$$\mathcal{G}_{enc}\left(\mathbf{x}, \mathcal{G}_{GENIE}, \mathcal{H}_{enc}(\boldsymbol{\mu}; \Phi)\right) = \mathbf{v}(\mathcal{G}_{GENIE})$$

Unlike the traditional *Hash Grid Encoding*, where the output depends directly on the query point $\mathbf{x}$, here the features are derived from nearby Gaussians. This is achieved by selecting the $N$ closest Gaussians to $\mathbf{x}$ using our RT-GPS algorithm (detailed in the following section). The final feature vector is computed as a weighted interpolation of features assigned to these Gaussians, using a modified Mahalanobis distance:

$$\mathbf{v}\left(\mathcal{G}_{GENIE}\right) = \sum_{i=1}^{k} w_i(\mathcal{G}_{GENIE}) \cdot \mathcal{H}_{\text{enc}}(\boldsymbol{\mu_i}; \boldsymbol{\Phi}),$$

$$w_i(\mathbf{x}) = \begin{cases} \exp\left(-\frac{1}{2}(\mathbf{x} - \boldsymbol{\mu}_i)\boldsymbol{\Sigma}_i^{-1}(\mathbf{x} - \boldsymbol{\mu}_i)\right), & \text{if } i \in N \\ 0, & \text{otherwise} \end{cases},$$

where $w_i(\mathbf{x})$ denotes the interpolation weight, $k$ is the maximum number of nearest neighbors considered, and $\boldsymbol{\Sigma}_i = \text{diag}(\exp(\mathbf{c}_i)) \in \mathbb{R}^{3 \times 3}$ is the diagonal covariance matrix parameterized for numerical stability via the log-domain vector $\mathbf{c}_i \in \mathbb{R}^3$. The features $\mathcal{H}_{\text{enc}}(\boldsymbol{\mu}_i; \boldsymbol{\Phi})$ are generated from a trainable hash-grid encoding and depend on the current Gaussian parameters.

During training, both the hash grid parameters $\Phi$ and the Gaussian means $\boldsymbol{\mu}_i$ are updated jointly, allowing the Gaussians to explore the multi-LoD feature field and shape the encoding. At inference time, the Gaussians' positions are frozen but can be manipulated. Since the interpolation scheme remains unchanged, any modification to Gaussian parameters leads to modifications in the output renderings.

**Ray-Traced Gaussian Proximity Search** Since nearest neighbor search is the bottleneck of our method, we employ an efficient approximation scheme. We observe that excluding certain Gaussians from the neighborhood set $N$ introduces only a bounded error $\varepsilon$ in the interpolated feature vector $\mathbf{v}(\mathcal{G}_{GENIE})$, which we formally derive in the Appendix A.2.

This observation serves as the starting point for our approximated nearest neighbor finding method: Ray-Traced Gaussian Proximity Search (RT-GPS). RT-GPS restricts nearest neighbor candidates to Gaussians whose confidence ellipsoids (defined by a quantile parameter $Q$) contain the query point $\mathbf{x}$. This reduces neighbor search to a point-in-sphere test, which we approximate using circumscribed icosahedrons for efficient computation.

RT-GPS method extends the RT-kNNS algorithm (Nagarajan et al., 2023), which finds neighbors within a fixed radius by checking if query points lie inside expanded spheres. We adapt this by assigning each Gaussian an individual radius based on its covariance,

$$r_i = Q \cdot \max\left\{\lambda \in \sigma(\boldsymbol{\Sigma}_i)\right\},$$

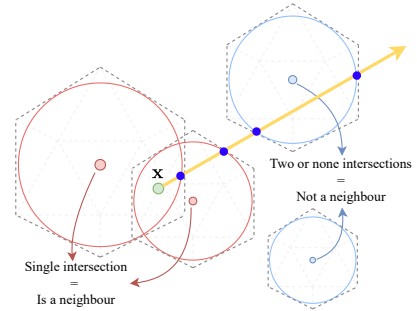

Figure 5: **The RT-GPS working principle.** A light ray passing through the scene is illustrated, along with its intersections with the icosahedrons. The figure highlights which Gaussians are considered neighbors and which are excluded.

where $\sigma(\boldsymbol{\Sigma}_i)$ is the set of eigenvalues of the Gaussian's covariance matrix and $Q$ is a configurable quantile. This ensures we only consider Gaussians whose confidence ellipsoids are likely to influence the feature at $\mathbf{x}$.

Following (Nagarajan et al., 2023), we trace rays from $\mathbf{x}$ and collect Gaussians whose confidence spheres intersect the ray exactly once (Figure 5). A sorted buffer maintains the $k$ closest candidates based on mean distance, and in case of overflow, the set is refined by rerunning the traversal with retained neighbors.

To limit traversal cost, we set the maximum ray distance to

$$t_{\max} = 2 \cdot \max_{i=1}^{n}\left\{Q \cdot \max\left\{\lambda \in \sigma(\boldsymbol{\Sigma}_i)\right\}\right\},$$

which guarantees that no significant Gaussians are skipped.

**Pruning and Densification** For densification, we adopt the strategy proposed in (Xu et al., 2022), defined as:

Figure 6: **Example edits on real-world scenes.** From left to right: (1) Physics-based simulation in the *Garden* scene from *Mip-NeRF 360*, showing an object falling onto a tilted table and bouncing off. (2) Physics simulation in the *kitchen* scene from *Mip-NeRF 360*, where a force is applied to deform a plasticine dozer.

$$\alpha_i = 1 - \exp(-\sigma_i \Delta_i), \quad j = \arg\max_i \alpha_i,$$

where $\alpha_i$ is the opacity at sample $i$ along a ray, $\Delta_i$ is the spacing between samples, and $j$ is the index of the maximum-opacity point. A new Gaussian is added at the shading location with the highest opacity only if its distance to existing closest Gaussians exceeds a predefined spatial threshold $\tau_s$, and its opacity value is above an opacity threshold $\tau_\alpha$. Unlike (Xu et al., 2022), we initialize features by sampling from the hash grid, rather than nearby shading information, ensuring better alignment with the feature field.

For pruning, we maintain a confidence vector $\mathbf{c} = [c_1, \ldots, c_n]$ with $c_i \in [0, 1]$. At each iteration, all values decay by a factor $\lambda_d < 1$, while Gaussians selected as neighbors are incremented by a growth factor $\lambda_g > 1$:

$$c_i \leftarrow \begin{cases} \min(1, \lambda_g \cdot c_i), & \text{if } i \in N(\mu, \Sigma) \\ \max(0, \lambda_d \cdot c_i), & \text{otherwise} \end{cases}.$$

Gaussians with $c_i < \tau$ are periodically.

**Editing** Thanks to the feature encoding in Splash Grid Encoding, we regularize the model's latent space around the spatial configuration of Gaussian primitives. This alignment allows edits to be performed directly in the coordinate space of Gaussians, effectively making spatial transformations equivalent to latent-space manipulations. In particular, modifying the means of the Gaussians enables localized scene edits that are instantly reflected in the rendered output.

Gaussians can be manipulated either individually or indirectly through mesh parametrization. In the latter case, we export the Gaussians as a triangle soup by projecting their two principal covariance directions onto triangle faces. Following the reparameterization strategy introduced in GaMeS (Waczyńska et al., 2024), we associate these triangles with mesh surfaces, ensuring that Gaussian components move consistently with mesh deformations.

All edits are applied in real time, with immediate visual feedback. Since the latent feature space is directly tied to Gaussian positions and attributes, the edits require no further fine-tuning or postprocessing, making them persistent and semantically meaningful.

## 5 EXPERIMENTS

We design our experiments to demonstrate that GENIE maintains the reconstruction quality of state-of-the-art (SOTA) methods while enabling complex object modifications.

**Datasets** Following prior work, we evaluate on the *NeRF-Synthetic* dataset (Mildenhall et al., 2020), which contains eight synthetic scenes with diverse geometry, texture, and specular properties. Existing methods (Govindarajan et al., 2024; Xu et al., 2022; Wang et al., 2023b) typically operate in bounded regions and do not support unbounded scenes. In contrast, GENIE is the first editable NeRF model trained on the challenging *Mip-NeRF 360* dataset (Barron et al., 2022), comprising five outdoor and four indoor real-world 360°scenes.

To further demonstrate editing capabilities, we include the *fox* scene from Instant-NGP (Müller et al., 2022), and introduce a custom set of 3D assets with deformable and articulated objects, enabling dynamic scene editing and physical interaction.

**Baselines** We compare GENIE against both static NeRF-based and editable point-based/Gaussian-based representations. For static radiance field models, we consider NeRF (Mildenhall et al., 2020), Nerfacto (Tancik et al., 2023), VolSDF (Yariv et al., 2021), ENVIDR (Liang et al., 2023), Plenoxels (Fridovich-Keil et al., 2022), GS, LagHash (Govindarajan et al., 2024), Mip-NeRF 360 (Barron et al., 2022), Instant-NGP (Müller et al., 2022), which are known for their high reconstruction quality, but lack support for scene editing.

| | Chair | Drums | Lego | Mic | Materials | Ship | Hotdog | Ficus |
|---|---|---|---|---|---|---|---|---|
| Static | | | | | | | | |
| NeRF | 33.00 | 25.01 | 32.54 | 32.91 | 29.62 | 28.65 | 36.18 | 30.13 |
| Nerfacto | 27.81 | 17.96 | 21.57 | 24.97 | 20.35 | 19.86 | 30.14 | 21.91 |
| VolSDF | 30.57 | 20.43 | 29.46 | 30.53 | 29.13 | 25.51 | 35.11 | 22.91 |
| ENVIDR | 31.22 | 22.99 | 29.55 | 32.17 | 29.52 | 21.57 | 31.44 | 26.60 |
| Plenoxels | 33.98 | 25.35 | 34.10 | 33.26 | 29.14 | 29.62 | 36.43 | 31.83 |
| GS | **35.82** | **26.17** | **35.69** | 35.34 | **30.00** | 30.87 | **37.67** | **34.83** |
| LagHash | 35.66. | 25.68 | 35.49 | **36.71** | 29.60 | **30.88** | 37.30 | 33.83 |
| Editable | | | | | | | | |
| RIP-NeRF | **34.84** | 24.89 | 33.41 | 34.19 | 28.31 | **30.65** | 35.96 | 32.23 |
| GENIE | 34.67 | **25.57** | 33.84 | **34.56** | 29.43 | 29.35 | **36.45** | **33.23** |

Table 1: Quantitative comparisons (PSNR) on a NeRF-Synthetic dataset showing that GENIE gives comparable results with other models.

For editable models we compare ourselves with RIP-NeRF (Wang et al., 2023b) and NeurlaEditor (Chen et al., 2023). We select these baselines to demonstrate that GENIE not only achieves comparable or superior reconstruction quality to SOTA methods, but also introduces significantly more expressive and flexible editing capabilities. In addition, we present qualitative visual comparisons of physics simulations generated with PhysGaussian (Xie et al., 2024) and GASP (Borycki et al., 2024), as shown in Fig. 7.

**Quantitative Results** We present quantitative results on the *NeRF-Synthetic* dataset in Table 1. As shown, GENIE achieves reconstruction quality comparable to SOTA non-editable methods. Among these, 3DGS performs best in terms of pure reconstruction fidelity. In the editable category, our method significantly outperforms RIP-NeRF in six out of eight scenes, and performs on par in the remaining two.

| | Outdoor scenes | | | | | Indoor scenes | | | |
|---|---|---|---|---|---|---|---|---|---|
| | bicycle | flowers | garden | stump | treehill | room | counter | kitchen | bonsai |
| Static | | | | | | | | | |
| INGP | 22.17 | 20.65 | 25.07 | 23.47 | 22.37 | 29.69 | 26.69 | 29.48 | 30.69 |
| Nerfacto | 17.86 | 17.79 | 20.82 | 20.48 | 16.72 | 24.22 | 23.59 | 23.20 | 21.55 |
| Mip-NeRF | 24.37 | **21.73** | 26.98 | 26.40 | **22.87** | **31.63** | **29.55** | **32.23** | **33.46** |
| GS | **25.25** | 21.52 | **27.41** | **26.55** | 22.49 | 30.63 | 28.70 | 30.32 | 31.98 |
| Editable | | | | | | | | | |
| GENIE | 19.47 | 18.14 | 22.29 | 20.04 | 16.34 | 28.57 | 24.98 | 25.69 | 25.94 |

Table 2: The quantitative comparisons of reconstruction capability (PSNR) on *Mip-NeRF 360* dataset.

For real-world scenes, we report PSNR on the *Mip-NeRF 360* dataset in Table 2, where Mip-NeRF achieves the highest reconstruction quality.

**Qualitative Results** For the qualitative comparison, we utilized results reported by (Chen et al., 2023), where objects from the *NeRF-Synthetic* dataset were modified to evaluate editing performance. The visual quality of the edits was assessed across different methods. We observe that GENIE outperforms other approaches in the task of zero-shot editing, producing visibly higher-quality results. In particular, it more accurately reconstructs lighting reflections in the *Mic* scene, handles stretching in *Drums* more naturally, and introduces fewer artifacts in shadowed regions of *Hotdog* and *Lego*. The comparison is shown in Figure 8. Point-NeRF appears in the comparison as it was adapted to support editing by the authors of the NeuralEditor method.

**Physic-based Editing** We conducted a series of physics-based simulations in Blender (Community, 2018) using the mesh-driven editing mechanism described earlier. These experiments span both synthetic and real-world datasets and include diverse physical phenomena such as rigid body

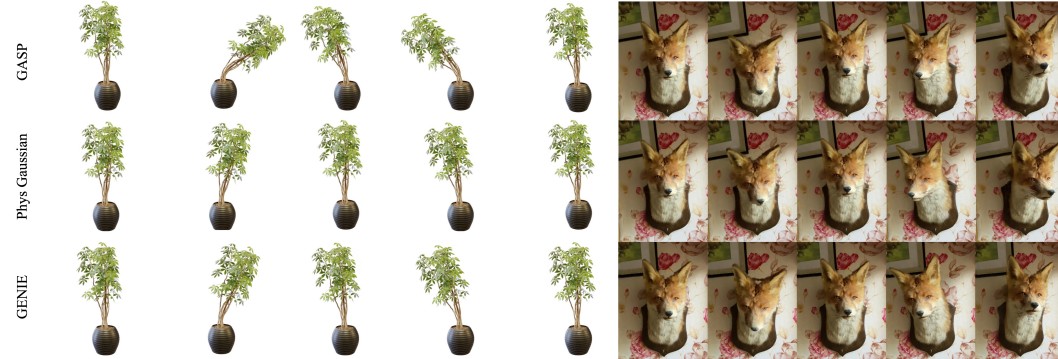

Figure 7: **Comparison of physics simulations with Gaussian Splatting methods.** From left to right: (1) wind simulation on the *Ficus* plant from the *NeRF-Synthetic* dataset, (2) particle impact simulation on the *Fox* head from the *Instant-NGP* dataset. Results shown for PhysGaussian (Xie et al., 2024) and GASP (Borycki et al., 2024).

dynamics, soft body deformation, and cloth simulation. In these scenarios, deformations of the driving mesh were used to update the corresponding Gaussian components in real time, enabling seamless integration of physical interactions into the scene. In addition, we performed simulations following PhysGaussian (Xie et al., 2024) and compared GENIE qualitatively against both Phys-Gaussian and GASP (Borycki et al., 2024).

The results of these simulations are illustrated in Figures 2, 3, 6, and 7. These visualizations demonstrate that GENIE produces realistic and physically plausible edits across a wide range of scenarios. Whether simulating leaves falling from a plant, squashing a soft object, or draping cloth over complex geometry, our method maintains high rendering fidelity while enabling expressive and controllable scene manipulation. This highlights the potential of GE-NIE as a flexible framework for neural scene editing driven by physical interactions.

Figure 8: **Qualitative comparison.** Results shown on the *NeRF-Synthetic* dataset. Modified objects are in the top row. Each row compares reconstruction quality across different methods. Our results are added to those reported by (Chen et al., 2023).

## 6    CONCLUSIONS

In this work, we introduced GENIE, a Gaussian-based conditioning technique for NeRF systems that enables dynamic and physics-driven editing. Our method conditions a NeRF network on jointly trained Gaussians that serve as spatial feature carriers. Editing can be performed either manually, through direct manipulation of Gaussians, or automatically, by coupling them with deformable meshes to enable physics-based interactions. We demonstrated the capabilities of our system across a range of scenarios, highlighting its usability, versatility, and adaptability. GENIE can be seamlessly integrated into new simulation environments, offering a promising path toward physically interactive neural scene representations.

**Limitations** The detail reconstruction quality in our system depends on the spatial density of Gaussians. Sparse regions may lose fine details, posing challenges in large or open scenes where maintaining uniform density is difficult.

## 7 Reproducibility Statement

We are committed to ensuring that our work is fully reproducible. To this end, we provide the following:

- Code and Environment: The complete source code for our method, including training, editing, and rendering pipelines, is available in our GitHub repository (link anonymized for review https://anonymous.4open.science/r/genie-B60F/README.md). The repository contains a README.md with detailed installation instructions. We support both Docker-based deployment (recommended) and manual installation. All dependencies and version requirements are explicitly documented.

- Hardware and Software: All experiments were run on a single NVIDIA RTX 3090 GPU (24 GB). We rely on CUDA, OptiX (7.6 or 8.1.0), and PyTorch as the main libraries. We also provide a Docker container that reproduces our environment without requiring manual configuration.

- Datasets: For synthetic experiments, we use publicly available datasets such as NeRF-Synthetic and Mip-NeRF 360. For real-world experiments, we follow the Nerfstudio data format. We will release all preprocessed datasets and corresponding configuration files.

- Training and Hyperparameters: We provide configuration files for every experiment reported in the paper. These include all hyperparameters such as learning rates, thresholds, and training schedules. Commands for training, evaluation, and rendering are provided in the README.

- Additional remarks: Reproducing Blender-based animations requires installing Blender, which we describe in detail. Rendering physics simulations additionally requires mesh export. We provide example scripts and workflows in our repository.

Taken together, these resources ensure that our results can be reliably reproduced by independent researchers.

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

# A  APPENDIX

This appendix provides additional insights and supporting material for our method. We begin with implementation details covering initialization, training schedules, and pruning strategies. Next, we give a formal justification of the $k$-nearest neighbor approximation used in Ray-Traced Gaussian Proximity Search, showing that distant Gaussians can be safely ignored with bounded error. We then present an extensive ablation study to analyze the impact of key components in our system. Next, we include rendering speed comparisons with existing methods, highlighting the trade-off between editability and performance. We also provide extended qualitative results to showcase the generalization of our approach across various scenes. Finally, we present and discuss representative failure cases to inform future research directions and reveal current limitations of our method.

## A.1  IMPLEMENTATION DETAILS

To reduce computation, we fix the rotation matrix $R$ to identity and restrict the covariance $\Sigma$ to a diagonal form, avoiding costly matrix inversions. The log-diagonal of $\Sigma$ is initialized to 0.0001.

For Splash Grid Encoding, we use quantiles $Q \in [1, 3]$ and select 16–32 nearest Gaussians per query. Densification runs periodically from early training until midpoint, adding up to 10,000 Gaussians per cycle. We use an opacity threshold $\tau_\alpha = 0.5$ and spatial threshold $\tau_s = 0.001$.

Pruning is performed every 1,000 steps. Confidence values decay via $\lambda_d = 0.001$ and increase via $\lambda_g = 0.01$. Gaussians with confidence $< \tau = 0.1$ are removed. Models are trained for 20,000 steps.

For initialization on the *NeRF-Synthetic* dataset, we used Gaussians generated by the LagHash method. For the *Mip-NeRF 360* dataset, we initialized GENIE using structure-from-motion reconstructions from COLMAP (Schönberger & Frahm, 2016), and further augmented the scene with an additional 1M points distributed along the scene boundaries to improve background reconstruction. For our custom assets, we initialized the Gaussians using the mesh vertices. For physics simulations, we used meshes generated with Permuto-SDF (Rosu & Behnke, 2023) but also simple cage meshes for real scenes. All experiments were run on a single NVIDIA RTX 3090 (24 GB) GPU.

## A.2 THEORETICAL MOTIVATION FOR RAY-TRACED GAUSSIAN PROXIMITY SEARCH APPROXIMATION

To justify the motivation behind our *Ray-Traced Gaussian Proximity Search*, let's first recall the formula for the interpolated feature vector $\mathbf{v}\left(\mathcal{G}_{GENIE}\right)$. To begin, let's note that for the $w_i(\mathbf{x})$ appearing in the formula we have:

$$w_i(\mathbf{x}) = \begin{cases} \exp\left(-\frac{1}{2} d_M^2\left(\mathbf{x}, \mathcal{N}\left(\boldsymbol{\mu}_i, \boldsymbol{\Sigma}_i\right)\right)\right), & \text{if } i \in N \\ 0, & \text{otherwise}, \end{cases}$$

where $d_M\left(\mathbf{x}, \mathcal{N}\left(\boldsymbol{\mu}_i, \boldsymbol{\Sigma}_i\right)\right)$ is the Mahalanobis distance of the point $\mathbf{x}$ from the normal distribution $\mathcal{N}\left(\boldsymbol{\mu}_i, \boldsymbol{\Sigma}_i\right)$. Let's fix $\mathbf{x} \in \mathbb{R}^3$ and $\varepsilon > 0$. Let's consider the subset $M \subseteq N$, such that for each $i \in M$ we have:

$$d_M\left(\mathbf{x}, \mathcal{N}\left(\boldsymbol{\mu}_i, \boldsymbol{\Sigma}_i\right)\right) > \sqrt{-2\ln\left(\frac{\varepsilon}{\sum\limits_{i \in M} |\mathbf{v}(\mathbf{x})_i|}\right)}$$

Then:

$$\left| \sum_{i \in M} w_i(\mathbf{x}) \cdot v(\mathbf{x})_i \right| =$$

$$= \left| \sum_{i \in M} e^{-\frac{1}{2} d_M^2(\mathbf{x}, \mathcal{N}(\boldsymbol{\mu}_i, \boldsymbol{\Sigma}_i))} \cdot v(\mathbf{x})_i \right| \le$$

$$\le \sum_{i \in M} \left| e^{-\frac{1}{2} d_M^2(\mathbf{x}, \mathcal{N}(\boldsymbol{\mu}_i, \boldsymbol{\Sigma}_i))} \right| \cdot |v(\mathbf{x})_i| =$$

$$= e^{-\frac{1}{2} d_M^2(\mathbf{x}, \mathcal{N}(\boldsymbol{\mu}_i, \boldsymbol{\Sigma}_i))} \cdot \sum_{i \in M} |\mathbf{v}(\mathbf{x})_i| <$$

$$< \frac{\varepsilon}{\sum\limits_{i \in M} |\mathbf{v}(\mathbf{x})_i|} \cdot \sum_{i \in M} |\mathbf{v}(\mathbf{x})_i| = \varepsilon$$

Thus:

$$\left| \sum_{i \in N} w_i(\mathbf{x}) \cdot v(\mathbf{x})_i - \sum_{i \in N \setminus M} w_i(\mathbf{x}) \cdot v(\mathbf{x})_i \right| =$$

$$= \left| \sum_{i \in M} w_i(\mathbf{x}) \cdot v(\mathbf{x})_i \right| < \varepsilon$$

from which we conclude that removing the nearest neighbors from the set $M$ from the formula for $\mathbf{v}\left(\mathcal{G}_{GENIE}\right)$ can alter the interpolated feature vector coordinate by no more than $\varepsilon$.

## A.3 ABLATION STUDY

To justify our design choices, we present an ablation study evaluating the impact of key components in our system. We analyze how performance is affected by the number of neighbors used in RT-GPS,

using Gaussian scales as radii in RT-GPS ($\Sigma$ in RT-GPS), the presence of Splash Grid Encoding, enabling densification and pruning, making Gaussian means learnable, and including an appearance embedding.

| $Q$ | # Neighbors | $\Sigma$ in RT-GPS | Splash Grid Encoding | Densification | Pruning | Learnable means | Appearance embedding | Chair | Drums | Lego | Mic | Materials | Ship | Hotdog | Ficus |
|---|---|---|---|---|---|---|---|---|---|---|---|---|---|---|---|
| | | | | | PSNR | | | | | | | | | | |
| GENIE 1.1 | 16 | ✗ | ✗ | ✗ | ✗ | ✗ | ✗ | 33.95 | 24.80 | 32.80 | 33.44 | 28.51 | 28.81 | 35.48 | 32.57 |
| GENIE 1.1 | 16 | ✗ | ✗ | ✗ | ✓ | ✗ | 32 | 29.27 | 23.47 | 25.15 | 28.72 | 21.94 | OOM | 30.19 | 27.92 |
| GENIE 1.1 | 16 | ✗ | ✗ | ✗ | ✓ | ✗ | ✗ | 33.92 | 24.84 | 32.42 | 33.26 | 28.26 | OOM | 34.98 | 31.87 |
| GENIE 1.1 | 16 | ✗ | ✗ | ✗ | ✗ | ✗ | 32 | 28.67 | 23.43 | 24.95 | 28.58 | 21.85 | OOM | 29.07 | 27.53 |
| GENIE 1.1 | 16 | ✗ | ✗ | ✗ | ✗ | ✗ | 10 | 30.59 | 23.69 | 27.84 | 29.03 | 24.30 | OOM | 31.52 | 27.97 |
| GENIE 1.1 | 16 | ✗ | ✗ | ✗ | ✗ | ✗ | ✗ | 33.11 | 24.89 | 32.67 | 32.57 | 28.43 | OOM | 35.26 | 32.11 |
| GENIE 1.1 | 16 | ✓ | ✗ | ✗ | ✗ | ✗ | ✗ | 34.10 | 24.94 | 33.02 | 33.82 | 21.16 | 28.91 | 35.71 | 32.07 |
| GENIE 1.1 | 16 | ✓ | ✗ | ✗ | ✓ | ✗ | ✗ | 34.12 | 24.97 | 33.00 | OOM | 28.61 | OOM | 35.66 | 32.11 |
| GENIE 1.1 | 16 | ✓ | ✗ | ✓ | ✓ | ✓ | ✗ | 32.92 | 25.08 | 32.98 | 22.74 | 28.15 | 28.23 | 35.54 | 32.43 |
| GENIE 1.1 | 16 | ✓ | ✓ | ✗ | ✗ | ✗ | ✗ | 34.20 | 25.15 | 33.09 | 33.88 | 29.03 | 29.43 | 35.89 | 32.53 |
| GENIE 1.1 | 16 | ✓ | ✓ | ✗ | ✗ | 5k | ✗ | 34.29 | 25.20 | 33.24 | 33.93 | 29.19 | 29.32 | 36.12 | 32.53 |
| GENIE 1.1 | 16 | ✓ | ✓ | ✗ | ✗ | 10k | ✗ | 34.34 | 25.24 | 33.38 | 34.03 | 29.18 | **29.45** | 36.18 | 32.54 |
| GENIE 2.0 | 32 | ✓ | ✓ | ✓ | ✓ | 10k | ✗ | **34.67** | **25.57** | **33.84** | **34.56** | **29.43** | 29.35 | **36.45** | **33.23** |

Table 3: Ablation study (PSNR) comparisons on a *NeRF-Synthetic* dataset showing that GENIE final system gives the best results. It can be observed that without Splash Grid Encoding system was sometimes giving the Our of Memory (OOM) errors.

### A.4 SPEED COMPARISONS

We compare the rendering performance of various methods in Table 4. For GENIE, we report results for different configurations based on the number of Gaussian components (in millions) and the number of nearest neighbors $k$ used to condition the NeRF. For example, "GENIE ∼1.1M, 32" denotes a model using approximately 1.1 million Gaussian components and $k = 32$ neighbors.

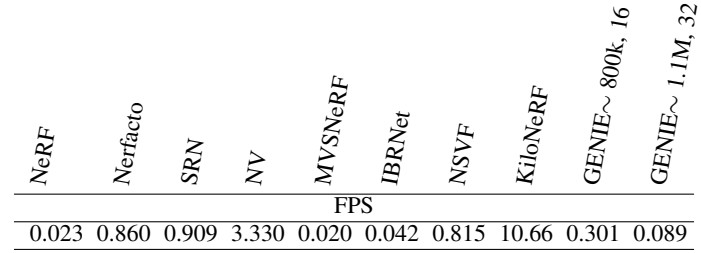

| NeRF | Nerfacto | SRN | NV | MVSNeRF | IBRNet | NSVF | KiloNeRF | GENIE∼ 800k, 16 | GENIE∼ 1.1M, 32 |
|---|---|---|---|---|---|---|---|---|---|
| | | | | FPS | | | | | |
| 0.023 | 0.860 | 0.909 | 3.330 | 0.020 | 0.042 | 0.815 | 10.66 | 0.301 | 0.089 |

Table 4: Rendering speed comparison on the *NeRF-Synthetic* dataset. Despite its editability features, GENIE achieves competitive inference speeds. Performance varies with the number of Gaussian components and neighbors used. For instance, "GENIE ∼1.1M, 32" refers to using approximately 1.1 million Gaussians and $k = 32$ neighbors in the weighted conditioning.

### A.5 FAILURE CASES

While our method performs well across a variety of scenes and tasks, it is not without limitations. In this section, we present representative failure cases that highlight scenarios where our approach struggles. The first failure mode occurs when the mesh model contains discontinuities caused by editing or undergoes excessive stretching. This can lead to visible holes or rendering artifacts in the

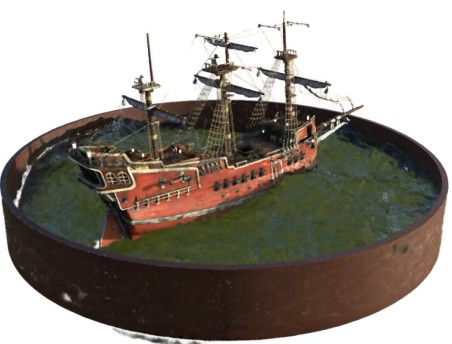

Figure 9: **Mesh discontinuity.** Mesh discontinuity during the editing causes holes in the edited model especially visible on the left side of the water basin.

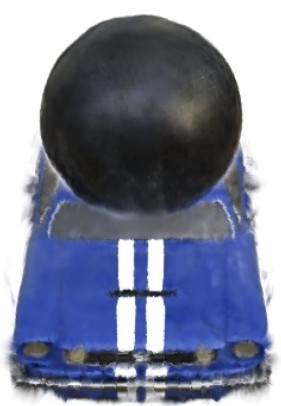

Figure 10: **Too few Gausses.** Too few Gausses during initialization and no densification causes network to have problems with proper reconstruction.

final output (see Figure 9). The second case arises when the number of Gaussians is insufficient during training and densification is disabled. In such situations, the network struggles to represent object boundaries accurately, leading to blurry or incomplete reconstructions (see Figure 10).

### A.6 EXTENDED RESULTS

In this section, we extend the results presented in Tables 1 and 2 of the main paper by additionally reporting SSIM and LPIPS metrics for both synthetic and real-world datasets.

| PSNR ↑ | | | | | | | |
|---|---|---|---|---|---|---|---|
| | Chair | Drums | Lego | Mic | Materials | Ship | Hotdog | Ficus |
| Static | | | | | | | | |
| NeRF | 33.00 | 25.01 | 32.54 | 32.91 | 29.62 | 28.65 | 36.18 | 30.13 |
| Nerfacto | 27.81 | 17.96 | 21.57 | 24.97 | 20.35 | 19.86 | 30.14 | 21.91 |
| VolSDF | 30.57 | 20.43 | 29.46 | 30.53 | 29.13 | 25.51 | 35.11 | 22.91 |
| ENVIDR | 31.22 | 22.99 | 29.55 | 32.17 | 29.52 | 21.57 | 31.44 | 26.60 |
| Plenoxels | 33.98 | 25.35 | 34.10 | 33.26 | 29.14 | 29.62 | 36.43 | 31.83 |
| GS | **35.82** | **26.17** | **35.69** | 35.34 | **30.00** | 30.87 | **37.67** | **34.83** |
| LagHash | 35.66. | 25.68 | 35.49 | **36.71** | 29.60 | **30.88** | 37.30 | 33.83 |
| Editable | | | | | | | | |
| RIP-NeRF | **34.84** | 24.89 | 33.41 | 34.19 | 28.31 | **30.65** | 35.96 | 32.23 |
| GENIE | 34.67 | **25.57** | **33.84** | **34.56** | **29.43** | 29.35 | **36.45** | **33.23** |

| SSIM ↑ | | | | | | | |
|---|---|---|---|---|---|---|---|
| | Chair | Drums | Lego | Mic | Materials | Ship | Hotdog | Ficus |
| Static | | | | | | | | |
| NeRF | 0.967 | 0.925 | 0.961 | 0.980 | 0.949 | 0.856 | 0.974 | 0.964 |
| Nerfacto | 0.951 | 0.835 | 0.893 | 0.959 | 0.771 | 0.797 | 0.951 | 0.915 |
| VolSDF | 0.949 | 0.893 | 0.951 | 0.969 | 0.954 | 0.842 | 0.972 | 0.929 |
| ENVIDR | 0.976 | 0.930 | 0.961 | 0.984 | **0.968** | 0.855 | 0.963 | **0.987** |
| Plenoxels | 0.977 | 0.933 | 0.975 | 0.985 | 0.949 | 0.890 | 0.980 | 0.976 |
| GS | **0.987** | **0.954** | **0.983** | **0.991** | 0.960 | **0.907** | **0.985** | **0.987** |
| LagHash | 0.984 | 0.934 | 0.978 | **0.991** | 0.947 | 0.892 | 0.981 | 0.981 |
| Editable | | | | | | | | |
| RIP-NeRF | 0.980 | 0.929 | **0.977** | 0.962 | 0.943 | **0.916** | 0.963 | **0.979** |
| GENIE | **0.981** | **0.934** | 0.973 | **0.987** | **0.950** | 0.880 | **0.979** | **0.979** |

| LPIPS ↓ | | | | | | | |
|---|---|---|---|---|---|---|---|
| | Chair | Drums | Lego | Mic | Materials | Ship | Hotdog | Ficus |
| Static | | | | | | | | |
| NeRF | 0.046 | 0.091 | 0.050 | 0.028 | 0.063 | 0.206 | 0.121 | 0.044 |
| Nerfacto | 0.056 | 0.197 | 0.112 | 0.075 | 0.405 | 0.218 | 0.029 | 0.112 |
| VolSDF | 0.056 | 0.119 | 0.054 | 0.191 | 0.048 | 0.191 | 0.043 | 0.068 |
| ENVIDR | 0.031 | 0.080 | 0.054 | 0.021 | 0.045 | 0.228 | 0.072 | **0.010** |
| Plenoxels | 0.031 | 0.067 | 0.028 | 0.015 | 0.057 | 0.134 | 0.037 | 0.026 |
| GS | **0.012** | **0.037** | **0.016** | **0.006** | **0.034** | **0.106** | **0.020** | 0.012 |
| LagHash | 0.024 | 0.083 | 0.027 | 0.015 | 0.070 | 0.139 | 0.036 | 0.049 |
| Editable | | | | | | | | |
| RIP-NeRF | - | - | - | - | - | - | - | - |
| GENIE | **0.013** | **0.060** | **0.016** | **0.005** | **0.037** | **0.110** | **0.022** | **0.015** |

Table 5: Quantitative comparisons (PSNR, SSIM, LPIPS) on a NeRF-Synthetic dataset showing that GENIE gives comparable results with other models.

PSNR ↑

| | Outdoor scenes | | | | | Indoor scenes | | | |
| | bicycle | flowers | garden | stump | treehill | room | counter | kitchen | bonsai |
|---|---|---|---|---|---|---|---|---|---|
| Static | | | | | | | | | |
| Plenoxels | 21.91 | 20.10 | 23.49 | 20.66 | 22.25 | 27.59 | 23.62 | 23.42 | 24.66 |
| INGP | 22.17 | 20.65 | 25.07 | 23.47 | 22.37 | 29.69 | 26.69 | 29.48 | 30.69 |
| Nerfacto | 17.86 | 17.79 | 20.82 | 20.48 | 16.72 | 24.22 | 23.59 | 23.20 | 21.55 |
| Mip-NeRF | 24.37 | **21.73** | 26.98 | 26.40 | **22.87** | **31.63** | **29.55** | **32.23** | **33.46** |
| GS | **25.25** | 21.52 | **27.41** | **26.55** | 22.49 | 30.63 | 28.70 | 30.32 | 31.98 |
| Editable | | | | | | | | | |
| GENIE | 19.47 | 18.14 | 22.29 | 20.04 | 16.34 | 28.57 | 24.98 | 25.69 | 25.94 |

SSIM ↑

| | Outdoor scenes | | | | | Indoor scenes | | | |
| | bicycle | flowers | garden | stump | treehill | room | counter | kitchen | bonsai |
|---|---|---|---|---|---|---|---|---|---|
| Static | | | | | | | | | |
| Plenoxels | 0.496 | 0.431 | 0.606 | 0.523 | 0.509 | 0.842 | 0.759 | 0.648 | 0.814 |
| INGP | 0.512 | 0.486 | 0.701 | 0.594 | 0.542 | 0.871 | 0.817 | 0.858 | 0.906 |
| Nerfacto | 0.548 | 0.495 | 0.559 | 0.657 | 0.591 | 0.815 | 0.773 | 0.692 | 0.728 |
| Mip-NeRF | 0.685 | 0.583 | 0.813 | 0.744 | 0.632 | 0.913 | 0.894 | 0.920 | **0.941** |
| GS | **0.771** | **0.605** | **0.868** | **0.775** | **0.638** | **0.914** | **0.905** | **0.922** | 0.938 |
| Editable | | | | | | | | | |
| GENIE | 0.381 | 0.329 | 0.465 | 0.424 | 0.404 | 0.834 | 0.718 | 0.655 | 0.736 |

LPIPS ↓

| | Outdoor scenes | | | | | Indoor scenes | | | |
| | bicycle | flowers | garden | stump | treehill | room | counter | kitchen | bonsai |
|---|---|---|---|---|---|---|---|---|---|
| Static | | | | | | | | | |
| Plenoxels | 0.506 | 0.521 | 0.386 | 0.503 | 0.540 | 0.419 | 0.441 | 0.447 | 0.398 |
| INGP | 0.446 | 0.441 | 0.257 | 0.421 | 0.450 | 0.261 | 0.306 | 0.195 | 0.205 |
| Nerfacto | 0.657 | 0.668 | 0.595 | 0.501 | 0.630 | 0.365 | 0.365 | 0.239 | 0.299 |
| Mip-NeRF | 0.301 | 0.344 | 0.170 | 0.261 | 0.339 | **0.211** | **0.204** | **0.127** | **0.176** |
| GS | **0.205** | **0.336** | **0.103** | **0.210** | **0.317** | 0.220 | **0.204** | 0.129 | 0.205 |
| Editable | | | | | | | | | |
| GENIE | 0.687 | 0.639 | 0.511 | 0.605 | 0.641 | 0.232 | 0.314 | 0.315 | 0.296 |

Table 6: The quantitative comparisons of reconstruction capability (PSNR, SSIM, LPIPS) on *Mip-NeRF 360* dataset.

