# OpenReview forum: "GENIE: Gaussian Encoding for Neural Radiance Field Interactive Editing"
_ICLR.cc/2026/Conference — ICLR 2026 Conference Withdrawn Submission_

### Official Review · Reviewer_cWxi · 2025-10-26

**Soundness:** 2
**Presentation:** 2
**Contribution:** 2
**Rating:** 2
**Confidence:** 4

**Summary:**

This paper aims to merge NeRF- and 3D Gaussian Splatting-style representations to improve editing capabilities. More specifically, the authors attempt to merge Gaussians into hash grids. Since hash grids are difficult to edit, the authors propose using Gaussians for feature interpolation instead. Because the Gaussians will not be directly used for rendering and will take the place of hash grid parameters, their colors are replaced with feature vectors. To sample features from these Gaussians, the authors propose a ray-traced Gaussian proximity search, along with pruning and densification stages. Throughout the experiments, the paper compares the proposed method with baseline methods and demonstrates physics-based editing results.

**Strengths:**

- The applications and necessity of an editable representation are well-presented with ample examples (Fig 3, 6, 7).
- Merging Gaussians in place of hash grids for editing is an interesting approach.

**Weaknesses:**

**Editability**: Simply using Gaussians instead of hash-based parameters is not sufficient to claim improved editability. There are numerous editing techniques, yet it is not well-documented how the proposed method addresses their limitations or where it provides tangible advantages. A broader comparison and clearer explanation of why this representation specifically improves editing would strengthen the claim.

**Motivation for Gaussians**: The rationale for integrating Gaussians into the representation for better NeRF-based editing remains unclear. If the goal is to combine the “best of both worlds” (NeRF and Gaussian Splatting), the paper should include comparisons with methods from both domains, discuss trade-offs, and demonstrate how the proposed approach leverages their respective strengths.

**Experimental performance**: Even if this representation is more editing-friendly, its quantitative performance currently lags behind the baselines, raising questions about its practical benefits.

**Questions:**

- The equations are not numbered. Adding numbers would improve clarity for both the authors and reviewers.
- The method section is somewhat difficult to understand. In Section 4, the paper states it uses Gaussians instead of grid parameters for "Splash grid encoding," noting that hash grids are not editing-friendly. However, if Gaussians are only used for feature interpolation, what do the grid parameters $\Psi$ represent? If the grid does use its own parameters, how does the paper resolve the contradiction that these parameters are "not good for editing" but are still being used?

---

### Official Review · Reviewer_7BsQ · 2025-10-31

**Soundness:** 2
**Presentation:** 3
**Contribution:** 2
**Rating:** 4
**Confidence:** 5

**Summary:**

This paper studies a specific type of 3D scene editing: scene deformation with human-guided inputs. The proposed method is to repurpose a Gaussian splatting representation to support NeRF-like volume rendering through interpolation-based point inquiry. The results show that the proposed method supports various deformation tasks.

**Strengths:**

- The method is straightforward and easy to understand.
- Code is provided in supplementary materials.

**Weaknesses:**

- All the quantitative results are just evaluating the reconstruction of static scenes, and the numbers of the proposed method are just comparable with the baselines.
    - Given that the proposed method can be regarded as an extension of 3DGS, it is certain that the performance should be comparable to 3DGS. However, in Tab.2, the performance dropped a lot compared with 3DGS.
    - NeuralEditor's NeRFSynthetic benchmark is used for qualitative evaluation in Fig.8. NeuralEditor has provided the ground truth of each deformed scene, and also reported the PSNR/SSIM metrics in their paper. However, this paper neither presents these metrics for its scene nor compares them with NeuralEditor's numbers.
    - Given that NeuralEditor's benchmark is the only benchmark for scene deformation, it is crucial to provide these numbers.
- The idea of this paper is a combination of multiple existing ideas.
    - The idea of interpolation-based point query and then an MLP is already the pattern of PointNeRF and NeuralEditor. Though this method is named "Splash Grid Encoding", it does not actually use any of the grid vertices, but is only dependent on the kNNs of the query point.
    - To find the kNNs for interpolation, the paper proposes RT-GPS. However, both PointNeRF and NeuralEditor propose their kNN finding methods. There is no discussion on how the proposed kNN method is better than the baselines.
- The view-dependent effects are confusing in this paper in both the method design and the results.
    - Given that the Gaussian parameters already consider view-dependency as SH factors, it is confusing why there is another "camera direction" input to the MLP F_GENIE in Fig.4.
    - In Fig.8, the bottom of the gong is completely unseen in training views, and this is why the baselines show strange colors. Even if the top of the gong is seen, the bottom may still be affected by view-dependent effects and have different colors. However, the proposed method shows a similar color to the top, which seems to assume it is view-independent.
- The actual rendering efficiency is not sufficient for the claimed "real-time" rendering. As shown in Appendix Tab.4, it takes at least 3.3 seconds to render a frame.

**Questions:**

- How is the training efficiency?
- Could you please provide the PSNR metrics for NeuralEditor's NeRFSynthetic benchmarks?
- What are the crucial idea differences between the proposed method and PointNeRF/NeuralEditor, specifically in the interpolation and kNN finding?
- While training-free for editing, NeuralEditor supports fine-tuning to improve the results if ground truth is given. Does the proposed method support this?
- As shown on NeuralEditor's website, in "Human-Guided Shape Deformation", row 1, column 2, NeuralEditor supports editing like "cutting a scene in the middle and rendering the two parts". Does the proposed method support this deformation?

---

### Official Review · Reviewer_g8yM · 2025-11-01

**Soundness:** 3
**Presentation:** 2
**Contribution:** 2
**Rating:** 4
**Confidence:** 3

**Summary:**

This paper proposes GENIE, a hybrid model that combines the high-quality rendering capabilities of NeRF with the editability of 3D Gaussian Splatting. The paper introduces the RT-GPS algorithm for fast nearest neighbor search and Splash Grid Encoding for multi-resolution feature encoding. The method supports real-time local editing and integration with physics-based simulation.

**Strengths:**

1. The overall structure is clear, with Figures 1 and 4 providing excellent overviews of the methodology.

2. Rich visualization of physics simulation results (Figures 2, 3, and 6).

**Weaknesses:**

1. The notation is sometimes inconsistent. For example, the definition of G_GENIE in Section 4 is overly complex, which compromises readability.

2. Some key concepts lack clear explanation. For instance, the use of "triangle soup" (Section 7) is not friendly to readers without a graphics background.

3. The baseline selection criteria for Tables 1 and 2 are not explained. Why are there no RIP-NeRF results on the Mip-NeRF 360 dataset?

4. The number of comparison methods is insufficient.

5. Figure 8 has low visual quality with text labels that are too small.

6. The experiments are insufficient in scope.

7. Table 2 shows that GENIE's PSNR is 5-7 dB lower than Mip-NeRF on Mip-NeRF 360, with even larger gaps in SSIM and LPIPS metrics. While the authors attribute this to Gaussian density, there lacks systematic analysis. The authors need to strengthen their justification.

8. There is no comparison of training time.

**Questions:**

1. What is the speedup ratio of RT-GPS compared to brute-force nearest neighbor search? Can you provide detailed time complexity analysis and experimental data?

2. The ablation study in Table 3 shows that removing Splash Grid Encoding causes OOM in some scenes (Ship, Mic). Does this indicate a design flaw in this module?

3. How do you ensure view consistency of rendering results after physics simulation? Are there artifacts from multiple viewpoints?

4. Is the method applicable to dynamic scenes or time-varying illumination?

---

### Official Review · Reviewer_oxnh · 2025-11-03

**Soundness:** 2
**Presentation:** 3
**Contribution:** 2
**Rating:** 4
**Confidence:** 3

**Summary:**

This paper introduces GENIE, a hybrid scene representation that marries the editability of 3D Gaussian Splatting with the photorealistic rendering of NeRFs to enable scene editing and physics-driven manipulation. Each Gaussian consists of a learnable feature embedding that conditions a NeRF; nearby Gaussians are selected efficiently with Ray-Traced Gaussian Proximity Search (RT-GPS), and the system supports densification/pruning to balance quality and speed. This design allows direct primitive manipulation or mesh-driven deformations to immediately reflect in rendered views without fine-tuning. Experiments on NeRF-Synthetic show reconstruction on par with strong baselines and competitive with 3DGS while outperforming editable baselines like RIP-NeRF on most scenes; on Mip-NeRF 360 the method remains editable though PSNR trails the best static models. The authors provide code, configs, and reproduction details.

**Strengths:**

1. The paper is well written and easy to follow
2. code, configs, hardware/software details, and workflows provided.
3. Novel hybrid architecture that meaningfully combines implicit and explicit representations, addressing a real limitation of pure NeRF methods (difficult editing)
4. Competitive reconstruction on NeRF-Synthetic and better than editable baselines like RIP-NeRF on most scenes; rendering speed competitive given editability.
5. Thorough ablations isolate the value of Splash Grid Encoding, RT-GPS, learnable means, and densification.

**Weaknesses:**

1. On Mip-NeRF 360, the proposed method achieves 19.47 PSNR vs 25.25 for Gaussian Splatting on bicycle scene - a 5.78 dB drop. Similar gaps appear across all outdoor scenes (Table 2). Even on NeRF-Synthetic, it underperforms GS (34.67 vs 35.82) and LagHash. This quality sacrifice is substantial and undermines the hybrid approach's value proposition.
2. While editing capabilities are a critical contribution of this work, there are no quantitative metrics for editing quality. Only qualitative comparison with PhysGaussian/GASP (Figure 7) are shown. Quantitative metrics, and potentially a user study would go a long way in emphasizing the editing capabilities of this method.
3. The rendering speed (Table 4) is considerably slower than prior works.
4. The system is extremely complex: there are many interacting components (RT-GPS with quantile Q, k neighbors, Splash Grid with multiple LoDs, pruning with confidence vectors, densification thresholds τ_s and τ_α). While ablation helps, the hyperparameter sensitivity and design choices aren't fully explored.

**Questions:**

1. The paper claims that the method is real-time (L42): but Table 4 shows that rendering is not real time. What does this claim mean?
2. Can you incorporate metrics for editing performance as well?
3. Why not use pure Gaussian Splatting editing? Given that Gaussian Splatting achieves better quality and faster rendering, what specific advantages justify the NeRF hybrid beyond avoiding "gaps" during super-resolution?
4. Can the method handle view dependent effects, such as specular highlights, as a result of editing?

---

### Note · Authors · 2025-12-02

I have read and agree with the venue's withdrawal policy on behalf of myself and my co-authors.